# Ins and Outs of Rocker Switch Mechanism in Major Facilitator Superfamily of Transporters

**DOI:** 10.3390/membranes13050462

**Published:** 2023-04-25

**Authors:** Stephanie Sauve, Joseph Williamson, Adithya Polasa, Mahmoud Moradi

**Affiliations:** Department of Chemistry and Biochemistry, University of Arkansas, Fayetteville, AR 72701, USA; ssauve@uark.edu (S.S.); jgw004@uark.edu (J.W.);

**Keywords:** major facilitator superfamily (MFS), membrane protein, rocker-switch mechanism, transporters, secondary membrane transporter proteins, alternating-access mechanism

## Abstract

The major facilitator superfamily (MFS) of transporters consists of three classes of membrane transporters: symporters, uniporters, and antiporters. Despite such diverse functions, MFS transporters are believed to undergo similar conformational changes within their distinct transport cycles, known as the rocker-switch mechanism. While the similarities between conformational changes are noteworthy, the differences are also important since they could potentially explain the distinct functions of symporters, uniporters, and antiporters of the MFS superfamily. We reviewed a variety of experimental and computational structural data on a select number of antiporters, symporters, and uniporters from the MFS family to compare the similarities and differences of the conformational dynamics of three different classes of transporters.

## 1. Introduction

Cells need specific concentrations of molecules and ions to be present on the inside and the outside of the cellular membrane for survival [1,2,3]. Movement in and out of the cell through the cellular membrane lipid bilayer can either be accomplished through passive diffusion or active transport. In passive diffusion, solutes move across their concentration gradient from a high concentration to a low concentration without requiring any additional energy to move to the other side [4,5,6]. On the other hand, in active transport, solutes move against their concentration gradients from a low concentration to a high concentration [7,8,9]. To do this, additional energy is required; usually, adenosine triphosphate (ATP) provides the energy needed to move the solutes across the opposing gradient [1,10,11]. However, the cellular membrane is semipermeable, meaning it allows for certain solutes to cross through the membrane to the inside or outside of the cell, but it does not allow for all solutes to cross on their own [12,13,14]. Therefore, in order to maintain adequate levels of solute concentrations on both sides of the membrane, transporters embedded in the membrane that help facilitate movement of molecules and ions across the membrane are essential [15,16,17,18,19,20,21,22]. In order to move the substrates across the membrane, the protein goes through a series of conformational changes. This review discusses the mechanism and corresponding conformational changes that are needed to facilitate this mechanism in the major facilitator superfamily (MFS).

MFS transporters are thought to be one of the oldest protein transporter families on the Earth and have existed for 3 billion years [23,24]. They are the largest group of secondary active membrane transporters in the cell and are found in almost all living organisms [20,24,25,26,27,28,29]. Therefore, they use an electrochemical gradient as a driving force for transport [22,30,31]. It is important to note that according to the Transporter Classification Database (TCDB, http://www.tcdb.org/ (accessed on 23 March 2023)) there is research that suggests the superfamily includes proteins whose main function is not to catalyze secondary transport. An example of such proteins are the major intrinsic proteins (MIPs) aquaporins and glycerol. Although these proteins contribute to the diversity found within this superfamily, they will not be discussed in this review. As a result of the diversity found within the superfamily due to it being one of the oldest protein transporter families, MFS transporters move a wide variety of substrates across the cellular membrane [23,24,25,27]. Some examples of the many substrates transported by the MFS include sugars (e.g., hexoses and pentoses), nitrite/nitrate, amino acids (e.g., histidine and lysine), oligopolypeptides (e.g., dipeptides and tripeptides), and organic cations [23,28,32,33,34,35].

Since the superfamily is so large with such wide diversity, it is easier to classify it into smaller groups. The superfamily can be divided into transporter types that are antiporters, symporters, and uniporters [28]. Antiporters move one or more substrates into the cell while simultaneously moving one or more substrates out of the cell or vice versa either with or against their concentration gradient. Symporters move two or more substrates in the same direction either with or against their gradients. Uniporters move one substrate at a time [36]. Although these transporters vary on how many substrates they carry and the direction in which they move them in, they undergo similar mechanisms within their distinct transport cycles, for example the rocker-switch mechanism. Although the similarities between these transporter types are important, their differences are also relevant since they could explain the distant functions of the superfamily. Therefore, we reviewed a variety of experimental and computational structural data on a select number of antiporters, symporters, and uniporters from the MFS family to compare the similarities and differences of the conformational dynamics of the three different classes of transporters.

In the following review, we discuss details of the MFS rocker-switch mechanism for each class of aforementioned transporters in depth. We focus on how the structure of the transporter type contributes to the mechanism of movement being carried out. Additionally, we highlight the importance of the formation and breakage of salt bridges in the movement of substrate from one side of the cellular membrane to the other. Each example discussed has a generalized mechanism for that specific transporter type along with a specific example of how salt bridges pertain to a transporter of interest of the transporter type. Such information has not been compiled to one review thus far.

### 1.1. Structure of MFS Transporters

The Transporter Classification Database (TCDB, http://www.tcdb.org (accessed on 23 March 2023)) identifies up to 106 different families within the major facilitator superfamily based on each transporters’ function and phylogeny. Though there have been hundreds of transporter proteins identified for the MFS, little progress has actually been made in terms of structural progress in the last two decades [37]. Despite the wide variety of mechanistic roles each one can perform, a similarity between all MFS transporters is their structural folding paradigm [19,36]. The canonical MFS-fold consists of a total organization of 12 transmembrane (TM) segments that further organize into 2 bundles (each with 6 TMs) where Bundle 1 at the N-terminal domain (NTD) contains TM 1–6, and Bundle 2 at the C-terminal domain contains TM 7–12 [29,38,39,40,41] (Figure 1). The organization at these domains is taken 1 step further as each bundle of 6 can be distinguished as 2 groups of 3 TM segments (1–3 and 4–6 or 7–9 and 10–12). Nevertheless, these domains join together to form a “⋀”—(inward facing) or “⋁”—(outward facing) shaped transporter showing NTD and CTD symmetry [22,42]. Although most MFS transporters contain 12 TM segments, transporters can have more or less than 12 TM segments, meaning that the number of TM segments in each bundle must be adjusted accordingly for these transporters [38,43].

The domain bundles are mainly connected by a long and flexible intracellular loop, though particular TMs of each bundle also interconnect with each other. The interconnections between TMs have been found to vary across different types of transporters [23,43,44]. Consistently, the substrate binding pocket is located at the interface between the bundles [29]. The structural details described above were identified using crystal structures of many MFS transporters, such as lactose symporter (LacY) [45], the glycerol-3-phosphate-phosphate antiporter (GlpT) of *Escherichia coli* [20], the oxalate transporter (OxlT) [46] of *Oxalobacter formigenes*, and the melibiose transporter (MelB) [47] of *Salmonella typhimurium*. It was also confirmed in these studies that structural differences between domains were largely due to the way substrates bind to the substrate binding pocket located between the two domains [48].

The domain bundles contain a structural similarity of perpendicular symmetry with respect to the plane of the membrane, also referred to as pseudo-two-fold symmetry [29,49]. This relationship has been identified as a 180${}^{\circ }$ rotation parallel to the plane of the membrane [40,50] (Figure 1). Another common similarity amongst MFS transporters are the inverted fold repeats found within all transporter folds [51]. These repeats can be explained in terms of the MFS bundles discussed throughout this paper, wherein one is present in the upright position, while the other is inverted by 180${}^{\circ }$ within the particular membrane. The term for this is “dual topology” which does not affect the membrane functionality as a homodimer [52].

Typically, MFS transporters have been generalized to express three conformational states as they work to transport relevant substrates in and out of a membrane. The three states are commonly referred to as the inward open/facing (IF) state where the binding site is in a space between the N- and C-domain on the cytoplasmic side; the occluded (OCC) state where the transporter is not open to either side of the membrane as a result of the helices that surround the substrate being compact, thus the substrate is unable to enter or leave the binding site; and the outward open/facing (OF) state where the binding space is on the extracellular side [21,29,36,53]. As one could imagine, the helices responsible for these movements would need to be either highly flexible or able to withstand breaking [54]. Research has identified particular helices responsible for motioning the substrate through the membrane gradient, referred to as the central cavity helices. Due to well-conserved glycine and/or proline residues with a tendency to bend in response to substrate binding, in partially and fully occluded (discussed later) states, these substrate gating helices have proven difficult to distinguish in the OF and IF states provided through crystallization [55].

### 1.2. The Rocker-Switch Mechanism

In 1990, Peter Mitchell opposed the previously standing moving carrier model that said transporters physically move with their substrates to the opposite side of the membrane with what he called the moving barrier model. In this model, he stated that the substrate binds to a specific point on the transporter, but the permeability barrier between that binding site and the two sides of the membrane could change so that the site was only accessible from one side of the membrane at a time [56]. Since then, mechanisms for alternating access models have been investigated. These methods state that transport proteins alternate between being open to the extracellular or intracellular side of the membrane so that the substrate cavity in the middle of the protein is only open to one side of the membrane at a time [21,22,30,57,58,59,60].

There are currently three models demonstrating the alternating access mechanism to describe how transporters move their substrates across the membrane [22] (Figure 2). They are called the rocking bundle model, the elevator model, and the rocker-switch model. The rocking-bundle mechanism suggests that the conformational change from inward open and outward open arise from tilting the helical bundles of one domain, while the second domain stays fixed [61,62]. On the other hand, the elevator model, which consists of one moving domain that slides through the membrane and one fixed domain, requires pushing an entire domain through the membrane [16,62,63]. The rocker-switch model is the third current model of the alternating access mechanism and stands in contrast to the rocking-bundle model and the elevator model.

The rocker-switch model explains the switching and thus alternating access between two conformational states, IF and OF open, seen in membrane transporters. The alternate access of the substrate binding site results from rocking the near symmetric N-domain and C-domain over a rotation axis that has the binding site at the interface of the domains [36,53]. However, there is research that suggests that although there is symmetry in the two domains, there can be non-symmetrical intra-domain movements that provide greater dynamic complexity [29]. As discussed previously, these domains are made up of helical bundles with helices 1–6 in the N-domain and helices 7–12 in the C-domain with the substrate binding site at the interface of the domains. The model states that the binding site is never open to both sides of the membrane at the same time [64]. Thus, once the rocking occurs, one side opens, while the other side closes, changing which side of the membrane the substrate binding site is open to allow substrates to move in and out of the cell (Figure 3). Therefore, since structure dictates function, a characteristic feature of rocker-switch proteins is the formation of two structurally similar domains that allow for the mechanism to be carried out [16,38]. Note that in the occluded conformation, the substrate binding site is not open to either the cytoplasm or periplasm [24,40,65] (Figure 4).

### 1.3. Importance of Salt Bridges for the MFS

Salt bridges are non-covalent interactions that occur between oppositely charged residues in close contact to each other and within hydrogen bonding distance that are able to form electrostatic interactions [66,67,68,69,70]. These non-covalent interactions can be relatively weak when looked at individually, but they have an additive effect, meaning that they can have great influence on the overall system as the number of salt bridges found in the system increases. Additionally, the overall strength of their interaction depends on the environment in which the salt bridges are found because water and various ions can impact salt bridge formation and stability [69,70,71,72,73]. Salt bridges can provide stability to a protein structure, influence the interactions of proteins with other biomolecules, and can play many roles in protein function [66,68,69]. Therefore, the possible formation of salt bridges is an important interaction to take into consideration when viewing protein structure, and the making and breaking of these interactions should be acknowledged when trying to determine stability of the various conformational states of a protein.

It is known that salt bridges are formed within and between the domains of MFS transporters [20,24,25,26,28,42,45,55,74,75,76,77,78,79,80,81,82]. It is thought that interdomain salt bridges are what hold the NTD and CTD together in some MFS transporters, such as Glpt and LacY [20,45]. Therefore, they are vital for maintaining protein structure and for allowing for movement of the domains relative to each other, which is vital for the mechanism of action. Additionally, interbundle salt bridges are typically found proximal to the substrate binding site in MFS transporters [24,32,36,55,83,84]. This indicates that their placement in MFS transporters may have a similar purpose and could be connected to substrate binding. Overall, salt bridges form an energetic barrier that can be overcome by substrate binding since the binding of the substrate can either break or form a salt bridge [24]. Therefore, it is worthwhile to look at the role of salt bridge formation and breakage in MFS transporters. In the examples discussed in this paper, we consider the role of salt bridges in the rocker-switch mechanism for each transporter type.

## 2. Significant Structural and Conformational Differences in the Rocker-Switch Mechanism across the Major Facilitator Superfamily of Transporters

As discussed previously, certain structural patterns remain constant across all transporter types in the MFS, such as the helical bundle domains that help facilitate the movement of substrates from one side of the membrane to the other. Although there are many similarities between how MFS transporters use the rocker-switch mechanism, there are also noteworthy differences. Keep in mind that the different transporter types can move varying amounts of substrates across the membrane. Because of this, they use the rocker-switch mechanism uniquely to accomplish their goal. In the subsequent sections of this review, we discuss the rocker-switch mechanism as it pertains to antiporters, symporters, and uniporters, respectively, with examples of use for each category. We also highlight the roles of salt bridges in regard to the mechanism for each example described.

### 2.1. The Rocker-Switch Mechanism for Antiporters

Antiporters are proteins embedded in the membrane that move two or more substrates across the cellular membrane in opposite directions via a series of substrate binding and conformational changes. It is known that antiporters in the MFS can use the rocker-switch mechanism [24,25,26,28,74,75] (Figure 5). Although the rocker-switch mechanism is identical across the three different types of transporters in the MFS, the mechanism is not entirely the same due to the variation in the number of substrates the transporter can move and the direction that it moves them in. For example, unlike the symporters that use the rocker-switch mechanism, antiporters face inward when substrate concentration is low until a substrate binds from the cytosolic side of the membrane unless regulatory action is being performed on the membrane protein [75]. When a substrate does bind, it changes the local pH and makes it less stable for protons to bind [75]. This is important because many antiporters use the proton-motive force (PMF) to drive their action of transporting substrates across the membrane. There are many antiporters that use the rocker-switch mechanism, including nitrate/nitrite transporters [28], glycerol-3-phosphate transporters [25,64,74], and oxolate/formate transporters [25] to name a few.

One example of an antiporter in the MFS is the glycerol-3-phosphate (GlpT) antiporter that uses the rocker-switch mechanism to move its substrates across the cellular membrane [74]. The glycerol-3-phosphate transporter (GlpT) moves one substrate, glycerol-3-phosphate, by using the inorganic phosphate concentration gradient on the opposite side of the membrane as its source of energy [26,64,85,86]. Binding of inorganic phosphate allows for the transition from the inward facing state to the outward facing state, allowing for movement to the opposite side [20,26,87]. To return to the inward facing state, glycerol-3-phosphate must bind [20,26,64,87,88]. Thus, a cycle of alternate binding occurs between the two substrates that changes which side of the membrane the binding site of the transporter protein is open to. Additionally, the GlpT antiporter is open to the cytosol side of the membrane with Helices 1 and 7 blocking substrate passage via aromatic residues that are contained on the helices that act as a blockade [25,26]. It has also been found that these helices are involved in the gating of the periplasmic side, while TM Helices H4, H5, H10, and H11 are more directly involved in gating of the cytoplasmic side [26]. Therefore, the gating ensures that substrates cannot freely flow through the transporter without the conformational shift occurring and correct substrate binding [29].

Mechanistically, it is thought that weak binding occurs by the substrate when binding residues are unprotonated [25,74]. Then, once protonation occurs there is greater interaction with the substrate pulling the helices closer and moving them, causing a conformational change [25,74]. Once the change is complete, the binding residues are deprotonated and therefore release the substrate on the opposite side of the membrane where it started [25,74]. This is consistent with the rocker-switch mechanism. The importance of protonation of specific residues has been seen in other MFS transporters as well, such as PiPT, a phosphate:H+ symporter, in which protonation of Asp 324 lowers the energy barrier for phosphate binding which provides a driving force for substrate translocation [29]. As a reminder, GlpT contains pseudo-twofold symmetry with the substrate binding site at the interface of the domains [20,26,87,89]. It is thought that because of the crystal structure of GlpT that the transitions between the inward facing and outward facing states is conducted by a ∼$10^{\circ }$ rigid-body rotation of the two halves [20,25,26,87].

Additionally, it has been shown that antiporters can form salt bridges that stabilize the different facing conformations, and the making and breaking of the salt bridges on substrate binding and release can also cause movement of the helices and thus conformational changes [25,26,28,75,78,82]. For example, Law et al. found that when inorganic phosphate binds loosely to GlpT and H165 is unprotonated that there is a strong salt bridge formed between D274 and R45 and between E299 and K46, with an additional weaker salt bridge formed between D274 and K46 [25]. However, when the histidine becomes protonated and the substrate binds tighter, the D274–K46 salt bridge is broken, and the E299–K46 salt bridge is weakened, while a new salt bridge is formed between E299 and R269 [25]. (Figure 6). The formation of this competing salt bridge assists with domain rotation [20,25,87] (Figure 6).

The structure of antiporters in the MFS can be complex, which may help antiporters distinguish between the large range of substrates that they can carry. For example, the OxlT transporter which binds many different substrates has long helical regions composed of Helices 4, 5, 10, and 11 that extend deep into the cytoplasm with charged residues [25]. These residues aide in ensuring substrate specificity is present despite being able to transport more than one substrate. Additionally, although binding residues for substrates are typically found on both the C-domain and the N-domain of interacting helices, some antiporters in the MFS contain binding residues on only one domain. This can been seen in OxlT and the oxolate/formate transporters [90].

### 2.2. The Rocker-Switch Mechanism for Symporters

The rocker-switch mechanism shown in Figure 7 is a well-known model for symporter conformational changes [91,92]. Symporters, which are membrane-integral proteins, are in charge of moving two or more molecules across the cell membrane at the same time (e.g., neurotransmitter transporters [91], sugar transporters [32,93], amino acid transporters [33,34], and nucleobase transporters [94,95]). According to many researchers, the symporter protein can exist in one of two major conformational states at any given time [32,33,34,64,91,92,93,94,95,96,97]. Two such states (Figure 2) are the outward-facing conformation and the inward-facing conformation [96,97], which are both maintained by various interactions between the protein and its substrates [96]. Proton-dependent oligopeptide transporter (POT) family proteins, also referred to as the PTR (peptide transport) family proteins, are additional members of the MFS symporters and are found in both Gram-positive and Gram-negative bacteria, yeast, plants, animals, and plants [42,64,98].

The POT family proteins mediate the absorption of a large variety of di- and tripeptides as well as peptide-like pharmacologically active compounds [64,76,99,100]. They use the inwardly directed proton (H+) electrochemical gradient to transport peptides across cell membranes in eukaryotes and prokaryotes [101,102]. Human peptide transporters 1 (PepT1), 2 (PepT2), and GkPOT are major POTs; PepT1 is mostly found in the small intestine, while PepT2 is found in the kidney, lungs, and central nervous system [77,102,103,104]. The human PepT protein contains a total of 14 TM helices, of which Helices H1–H12 adopt the general fold that was observed in other MFS transporters [20,41,45,105]. PepT1 and PepT2 encode the 14 common transmembrane (TM) helices, with both the N- and the C-terminal facing the cytoplasm [101,102,103,104,106].

The rocker-switch mechanism is associated with three steps in POT transporters (Figure 7). Step one is substrate binding-induced conformational change, step two involves protonation-induced conformational change, and step three involves release-induced conformational change [64,76,77,101,102,103,104,106,107,108]. A PepT crystal structure analysis revealed that the peptide-binding site is composed on the surfaces of both the N- and the C-terminal helix bundles, while the proton-binding site is situated close to the extracellular gate [76]. In the transport cycle’s first step, the peptide substrate binds to the POTs transporter in its outward-facing conformation. Then, protonation (proton binding) induces conformational changes after binding, in which TM helices undergo similar conformational changes [64,76,77,107,108]. This binding induces a conformational change that closes the transporter, and the transporter goes into an inward-facing position and exposes the peptide to the cytoplasm [77,107,108]. Consistent with experimental findings, a simulation study indicated that conformational changes in GkPOT are coupled to the transport of substrates and protons [77,107,108]. Finally, protons and peptides are released on the intracellular side of the membrane in this inward-facing conformation. A molecular dynamics simulation study of PepT [42,76,77] predicted that there are several salt bridges in the different conformations of PepT that keep the protein stable in its inward and outward shapes. Specifically, K127–E24, R25–E21, and R32–D316 salt bridges were consistently observed in all transport cycle conformations, whereas K127–E21 was predicted to form infrequently in inward and occluded conformations [42] (Figure 8). Mutating any of the residues involved in these salt bridges has previously been shown to abolish or significantly reduce transport [42,55].

Overall, similar rocker-switch mechanism steps are observed in other symporters, including sugar transporters, serotonin transporter (SERT), dopamine transporter (DAT), glutamate transporter, excitatory amino acid transporter (EAAT3), amino acid transporters, lactose permease (LacY), maltose transporter (MalFGK2), nucleobase transporters, such as uracil transporter (UraA) and xanthine transporter (XanQ), phosphate transporter (PiPT), and purine transporter (PUP1) [29,34,93,109,110,111,112,113,114,115,116,117].

### 2.3. The Rocker-Switch Mechanism for Uniporters

The uniporter transport system is likely the most identifiable transporter of the MFS transporters. Unlike symporters and antiporters, these systems function to move only a single substrate at a time either along or against the particular transporter gradient via facilitated diffusion [118] (Figure 9). Types of transporters that associate by facilitated diffusion are channel proteins and carrier proteins [119]. Glucose or GLUT transporters are a subset in the MFS TCBD Classification Database (TCBD ID 2.A.1, https://tcdb.org/search/result.php?tc=2.A.1 (accessed on 23 March 2023)) and actually among the first transporters to be functionally characterized with a substrate coordination map. GLUT1 is a particular ‘passive’ sugar porter (SP) uniporter found in a range of tissues essential for glucose transport into the brain [120]. The conformational transition from inside (without substrate) to outside the membrane was previously recorded as the slowest step of the cycle [121,122], likely due to the forming and breaking of the intracellular salt bridge network located at the ends of TM helices. Salt bridges within GLUT1 have not been recorded near the central cavity section for either outward or inward facing states [32,93,123,124,125].

A molecular dynamics study led by Dr. Etchebest studied aspects of the mechanics surrounding GLUT1 transfer and found two strongly connected salt bridge interactions that withstood nearly the entirety of the 1–1.5 microsecond MD simulations. The two main salt bridge interactions were reported as D240 with R232 (between intracellular Helices 2 and 3) and E247 with R212 (between intracellular Helices 3 and 1) (Figure 10). In turn, the intracellular portion of Helix 1 was found to have a strong connection to the intracellular portion of TM3 by residues E209 with R93 [97], as shown in Figure 10. These were not the only salt bridges identified through their simulations, but they establish a foundation for the underlying importance that salt bridges play in uniporter transportation.

A recent study on the human GLUT1 revealed that substrate binding causes the protein to “twist and tilt”, resulting in the formation of a hydrophilic pathway for glucose transport [97]. These uniporters are believed to undergo five main conformational states beginning as outward facing, outward-occluded, occluded (with or without substrate), inward-occluded, and inward facing (Figure 4). They have also been found to spontaneously reset themselves from the inward facing to outward facing conformation in the absence of substrate binding. Interestingly, this indicates that the energy barriers separating the two states must be relatively low [124].

First, in the initial step of the transport cycle, glucose is seen to bind to the outward-facing conformation of GLUT1. This binding induces a conformational change in the transporter, closing it and exposing glucose to the cytoplasmic side of the cell. It is believed that the transmembrane helices rotate relative to one another during this conformational change [93]. In this same study, cryo-EM was used to crystallize the structure of GLUT1 in the presence of glucose. The structure demonstrated the rotation of the transmembrane helices, consistent with previous biochemical and biophysical studies [97,126,127,128].

After binding to the outward-facing conformation of GLUT1, glucose molecules are transported across the membrane as the uniporter transitions to its first partial-occluded state (outward-occluded), (Figure 4). The helices typically responsible for these local TM movements are referred to as “gating helices” and are often apart of the central cavity helices. The responsibility of these helices is to ‘occlude’ the substrate from exiting during partial occlusion states [124]. In many cases, the responsible transmembranes interacting through the outward- and inward-facing state transitions are carried out by either TM4 or TM1 of the NTD with either TM10 or TM7 of the CTD (Figure 1) [24]. The fully occluded state has been hypothesized to be metastable for short-term occupation of the substrate during structural isomerization between the two partially occluded states [129].

Once the transporter has transitioned into the inward-facing state, the substrate will finally be released into the cytoplasm. This step involves an additional conformational change in which the transporter opens to expose the substrate-binding site on the extracellular side [123]. To allow the substrate release on the membrane’s internal side, the aforementioned transmembrane helices must move, often by breaking or due to high flexibility. This was supported by a study in which the structure of GLUT1 in its inward-facing conformation was determined using X-ray crystallography which also observed that the helices moved considerably more than in the outward-facing conformation [32].

## 3. Conclusions

MFS transporters have long been studied to provide insight into alternating-access mechanisms. Through this review and the included figures, we have described the structure of MFS transporters, competing alternating-access mechanisms, the importance of salt bridges for MFS transporters, and the rocker-switch mechanism as it pertains to antiporters, symporters, and uniporters in major facilitator superfamily transporters. We have shown that the overall structure of MFS transporters usually consists of 12 TM helices that are divided evenly into the CTD and NTD with an interdomain loop connecting the helical bundles. It is the movement of these two bundles that vary in the three current alternating-access mechanisms that are available, called the rocking-bundle model, the elevator model, and the rocker-switch model. In addition to generally describing the rocker-switch mechanism, we have shown that antiporters, symporters, and uniporters vary on the use of the rocker-switch mechanism according to how many substrates each transporter can move and the direction they move them in. However, the general structure and foundation of the mechanism remain consistent. In particular, we have gone through the mechanisms for GlpT, POT transporters, and GLUT1 as examples for the antiporters, symporters, and uniporters, respectively. We expect that the information provided in this review will provide insight on the rocker-switch mechanism and on the major facilitator superfamily as a whole. However, there is still much to learn about the MFS, and as available structures improve, more information can be gained. In particular, more research needs to be conducted on individual helical movement and domain movement to obtain a better understanding of the mechanisms employed by MFS and which alternating-access mechanisms best characterize their dynamics.

## Figures and Tables

**Figure 1 membranes-13-00462-f001:**
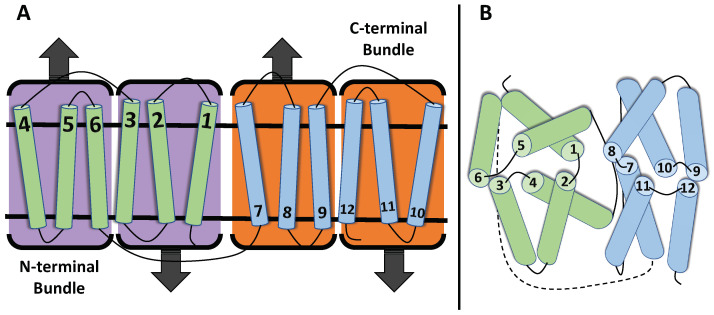
Schematic representation of the canonical MFS-fold transporter topology. (**A**) The canonical MFS topology is comprised of 12 TMs with a N and C orientation, forming two structurally similar six-helix bundles. The N-terminal bundle (TMs 1–6; green) and the C-terminal bundle (TMs 7–12; blue) are connected by a cytosolic loop, which can sometimes contain structural elements (black). Each of the bundles is made up from 3 TM structural-inverted repeats. (**B**) The first TM in each of the 3 TM repeats (TM1, TM4, TM7, and TM10) form the central cavity helices and often undergo local changes to bind and release the substrate during alternating access.

**Figure 2 membranes-13-00462-f002:**
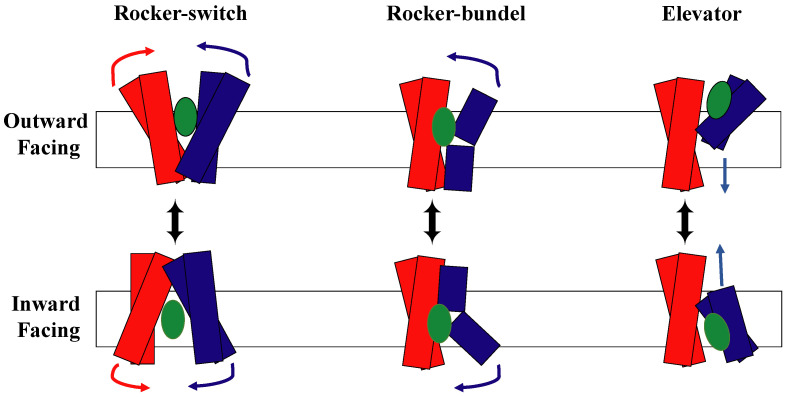
Schematic representation of the three types of current alternating access models with the substrate shown in green. The NTD is shown in blue and the CTD is shown in red.

**Figure 3 membranes-13-00462-f003:**
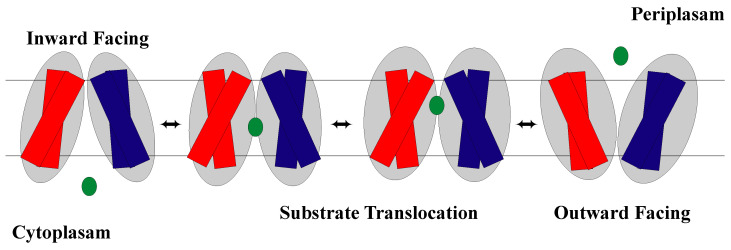
Schematic representation of the MFS facilitative diffusion of a substrate (green) across the cell membrane. The NTD is shown in blue and the CTD is shown in red.

**Figure 4 membranes-13-00462-f004:**
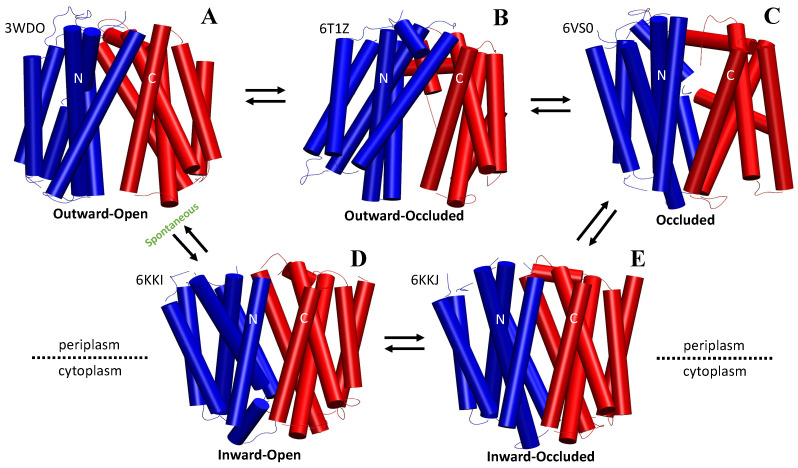
Crystallized depiction of the 5-state mechanism associated with many symporters and uniporters of the MFS that expand on the generalized cycle shown in Figure 3. Particular portions of some of the crystal structure of the proteins were excluded in order to capture the essence of the mechanism. (**A**) To display the outward-open state, the YajR (WT, PDB ID 3WDO) was used; residue IDs 387–455 were excluded. (**B**) The outward-occluded state, LmrP transporter from *L. lactis* (PDB ID 6T1Z), was used; residue IDs 501–603 were excluded. (**C**) The fully occluded state (with substrate), Protein B transporter of *E. coli* (PDB ID 6VS0), was used; residue IDs 501–680 were excluded. (**D**) The inward-occluded state was displayed using Drug:Proton Antiporter-1 (DHA1, PDB ID 6KKJ); residue IDs 401–403 were excluded. (**E**) The inward-open state was displayed using Drug:Proton Antiporter-1 (DHA1, PDB ID 6KKI).

**Figure 5 membranes-13-00462-f005:**
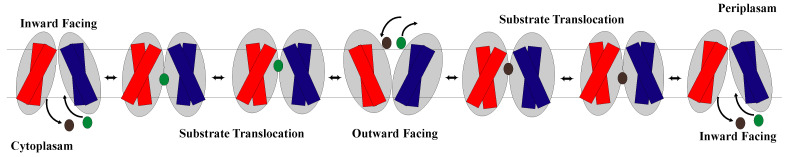
Schematic representation of the antiporter facilitating diffusion of two substrates (green and black) across the cell membrane with each substrate going in opposite directions. The NTD is shown in blue and the CTD is shown in red.

**Figure 6 membranes-13-00462-f006:**
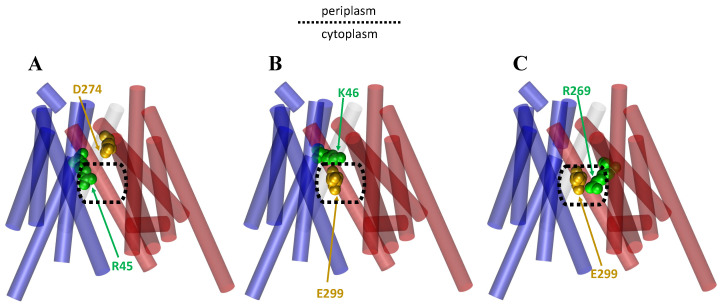
A cartoon representation of the GlpT transporter in its IF state based on its crystal structure (PDB ID: 1PW4), showing the salt bridges that form and break upon the binding of inorganic phosphate and the protonation of H165 in the Glpt transporter. The NTD is shown in blue and the CTD is shown in red. A dotted rectangle is provided to display the central cavities. The ‘gold’ and ‘green’ colors show the amino acid responsible for the negative and positive charge in the interaction, respectively: (**A**) the salt bridge interaction between D274 and R45 of the CTD and NTD, respectively; (**B**) the salt bridge interaction between E299 and K46 of the CTD and NTD, respectively; (**C**) the salt bridge interaction between E299 and R269 in both of the CTD.

**Figure 7 membranes-13-00462-f007:**
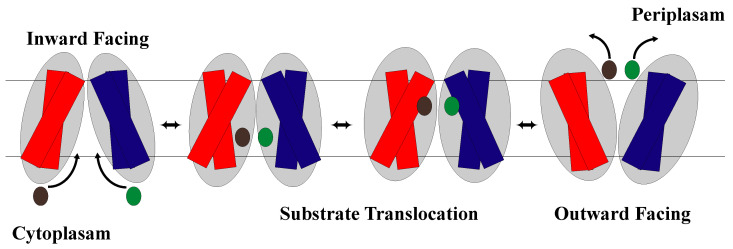
Schematic representation of the symporter facilitating diffusion of two substrates (green and black) across the cell membrane with each substrate going in the same direction. The NTD is shown in blue and the CTD is shown in red.

**Figure 8 membranes-13-00462-f008:**
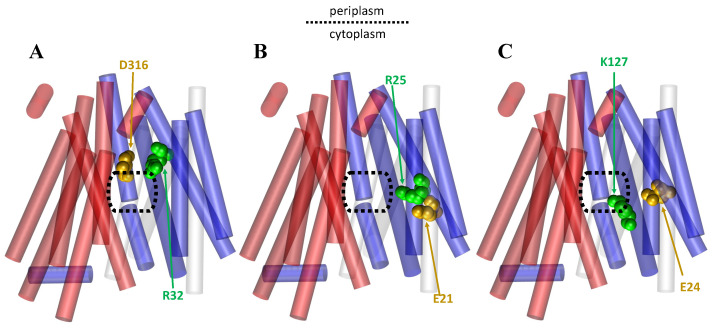
A cartoon representation of the PepT transporter in its IF state based on its crystal structure (PDB ID: 4UVM) showing the salt bridges that are consistently observed in all transport cycle conformations. The NTD is shown in blue and the CTD is shown in red. A dotted rectangle is provided to display the central cavities. The ‘gold’ and ‘green’ colors show the amino acid responsible for the negative and positive charge in the interaction, respectively: (**A**) the salt bridge interaction between D316 and R32 of the CTD and NTD, respectively; (**B**) the salt bridge interaction between E21 and R25, both of the NTD; (**C**) the salt bridge interaction between E24 and K127, both of the NTD.

**Figure 9 membranes-13-00462-f009:**
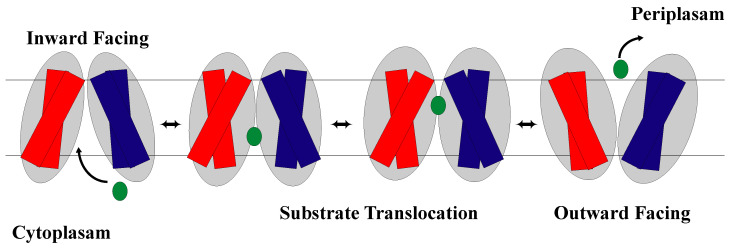
Schematic representation of the uniporter facilitating diffusion of one substrate across the cell membrane. The NTD is shown in blue and the CTD is shown in red.

**Figure 10 membranes-13-00462-f010:**
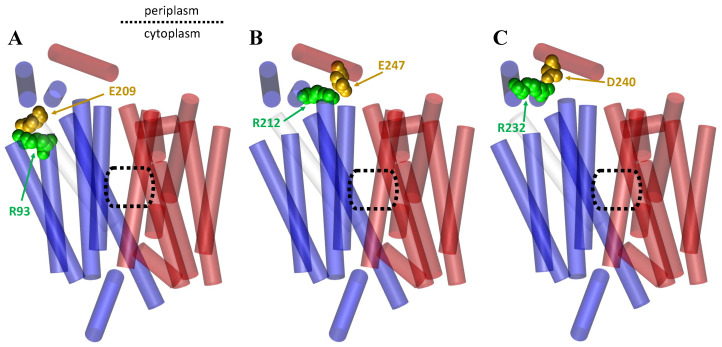
A cartoon representation of the GLUT1 transporter in its IF state based on its crystal structure (PDB ID: 4PYP), showing the salt bridges that are consistently observed in all transport cycle conformations. The NTD is shown in blue and the CTD is shown in red. A dotted rectangle is provided to display the central cavities. The ‘gold’ and ‘green’ colors show the amino acid responsible for the negative and positive charge in the interaction, respectively: (**A**) the salt bridge interaction between E209 and R93 of the CTD and NTD, respectively; (**B**) the salt bridge interaction between E247 and R212, both apart of the CTD; (**C**) the salt bridge interaction between D240 and R232, both apart of the CTD.

## Data Availability

No new data is reported in this review article.

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
