# Peer review of "Ins and Outs of Rocker Switch Mechanism in Major Facilitator Superfamily of Transporters"

_membranes, 2023, doi:10.3390/membranes13050462_

Round 1

Reviewer 1 Report

This review dedicated to the mechanisms of transport of molecules via the MSF family of transporters is a little superficial but it has the merit to gather information on the suggested implication of the salt bridges for the three sub-groups of transporters (Antiporter, Symporters, and Uniporters).

Nevertheless, it needs to be improved before publication. There are a lot of language errors and of description redundancy all along the text, especially in the introduction when describing the structures with the alternating opening towards the two sides of the membrane. The text really needs to be carefully examined and simplified to make the reading more fluid.

As the main interest of the review is in the description of the salt-bridges formation and dissociation governing the conformation changes leading to the transport of molecules, one would expect a discussion on their similarities and differences in the different sub-groups. It could be in the conclusion. Consider adding a few sentences on this point.

Below is a non-exhaustive list of modifications to be performed.

Line 36: Indicate the link for TCDB (https://tcdb.org/). It has to be given the first time it is cited

Line 38: "For example, the Major Intrinsic Protein (MIP) of aquaporins and glycerol." This sentence does not seem to be complete.

Line 48: a "s" is missing for "substrate"

Line 74: a dot is missing after [36]

Lines 86-88: the sentence is very difficult to understand, please reformulate

Line 89: the structural details described above

Lines 93-95: "It was also confirmed in these studies that structural differences between the two domains were largely due to the way substrates bind to the substrate binding pocket, which is located between the two domains "

In the legend of Figure 1, the colors are not correct, grey is yellow, and yellow is green…

There is a lot of redundancy in the description of the inward-to-outward movement and vice versa. Please simplify the 1.1 chapter

Line 128: "the substrate cavity", not "the substrate bound"

Figure 3 would be more appropriately placed after Figure 4.

It is very hard to see the opening on the structures, especially for the outward-open conformation. Could it be slightly turned? Or could you materialize the cavity?

Tell the orientation is the same as in Figure 2 or draw the membrane with the indication of the out and in zone. Be careful to adapt the legend and the figure. There is no a, b, c, d, e on the figure

Simplify the sentence on lines 209-213 as you just cited the transporter on line 207. It can be said: "The glycerol-3-phosphate (GlpT) moves one substrate, glycerol-3-phosphate, by using the inorganic phosphate concentration gradient on the opposite side of the membrane as the source of energy for its activity".

Line 296: omit "Other studies have shown that"

Several references are not properly cited, with names indicated as initials (see reference 51 as an example for which even the DOI is not correct). Verify all of them

Ref 120 does not exist any longer. Please update it

Author Response

Please see the point-by-point response.

Reviewer 2 Report

The article by Sauve et al. extensively reviews the mechanistic underpinnings of MFS transporters. The review is excellently compiled and covers all significant advances. I just missed one crucial study from Stroud's group about PiPT which was a very interesting one and should be included in the review (Pedersen et al. https://www.nature.com/articles/nature12042). 

volume pages533–536 (2013). I endorse the acceptance otherwise.

Author Response

Please see the point-by-point response.

Reviewer 3 Report

The manuscript by Sauve et al. is in general a well written review of the Rocker-Switch transport mechanism in the MFS superfamily. This review represents a good introduction to the MFS superfamily and its mechanisms of action.

In my opinion, the manuscript requires minor editing to be ready for publication:

1.    Page 2, lines 70-71: the text mentions that TCDB has identified up to 16 different families in MFS. This is an understandable, yet incorrect, statement. Currently, in MFS there are 106 different families (91 under TCID 2.A.1 and 15 with different TCIDs). Originally the MFS evolved as a family with multiple subfamilies in TCDB, but with time, subfamilies grew more divergent and it became clear that it was a superfamily. However, the IUBMB-approved policy of not changing TCIDs once established, prevented changing TCIDs. Thus, each subfamily or 4-component TCID (e.g., 2.A.1.1, 2.A.1.2, etc.) in MFS (TC: 2.A.1) became semantically a family.

2.    In the text, Figure 4 is cited (page 5, line 149) before Figure 3 (page 5, line 153). This is incorrect.

3.    Figure 3 has several problems. First, the legend of the figure talks about the MFS 5-state mechanism of symporters/uniporters a) to e), but the five structures of the figure are not labeled correspondingly, which would make the figure easier to understand. Yes, although there are titles such as Outward-open and Outward-occluded, adding a) to e) to each structure’s title will help non-experts understand (just like in figure 6). Second, the figure has no references to where the cytoplasm is located relative to the structures. Adding this point of reference will help non-expert readers understand. Third, the Inward-occluded state in the legend refers to structure 6KKI, but the figure has 6KKJ. Similarly, the inward-open state mentions structure 6KKJ in the legend, but the figure has 6KKI. Please correct these problems and your figure will be much easier to understand.

4.    Page 5, line 145, reads “…and helices 6-12 in the C-domain…”. Did you mean helices 7-12?

5.    Figures 6, 8 and 10. There is no reference to where the cytoplasm is located relative to the structures. Please, add it.

The paper is well written. However, I did notice a few minor typos:

1.    Page 1, line 1: At the beginning of the abstract, please add ‘The’ before ‘Major facilitator superfamily (MFS) …’

2.    Page 2, end of line 56: ‘We’ should be written in lower case because it is preceded by a comma.

3.    Page 9, end of line 277: the text ‘POT transporters(Fig.7.’ needs a space and a closing parenthesis: ‘POT transporters (Fig.7).’

4.    Page 11, line 311: Please, replace TCBD with TCDB twice in this line.

5.    Page 11, line 321: In this line, the authors talk about GluT1, but in the same page, lines 313 and 317 they use GLUT1. Please,  be consistent in the nomenclature for this particular transporter.

6.    Page 12, line 346: The open parenthesis in ‘(outward’ was never closed.

Author Response

Please see the point-by-point response.
